# A simple and versatile nickel platform for the generation of branched high molecular weight polyolefins

Tao Liang[1], Shabnam B. Goudari[1] & Changle Chen [1]*

The development of high-performance transition metal catalysts has long been a major driving force in academic and industrial polyolefin research. Late transition metal-based olefin polymerization catalysts possess many unique properties, such as the ability to generate variously branched polyolefins using only ethylene as the feedstock and the capability of incorporating polar functionalized comonomers without protecting agents. Here we report the synthesis and (co)polymerization studies of a simple but extremely versatile α-imino-ketone nickel system. This type of catalyst is easy to synthesize and modify, and it is thermally stable and highly active during ethylene polymerization without the addition of any cocatalysts. Despite the sterically open nature, these catalysts can generate branched Ultra-High-Molecular-Weight polyethylene and copolymerize ethylene with a series of polar comonomers. The versatility of this platform has been further demonstrated through the synthesis of a dinuclear nickel catalyst and the installation of an anchor for catalyst heterogenization.

[1] CAS Key Laboratory of Soft Matter Chemistry, Hefei National Laboratory for Physical Sciences at the Microscale, Department of Polymer Science and Engineering, University of Science and Technology of China, Hefei 230026, China. *email: changle@ustc.edu.cn

For more than half a century, transition metal-catalyzed olefin polymerization has been a global topic of intense research[1–4]. The metal nickel entered this field as a poison. Ziegler and coworkers demonstrated the famous nickel effect, in which ethylene was converted exclusively to 1-butene using aluminum alkyls in the presence of a small amount of nickel salt[5]. This discovery marks the starting point for the development of the Ziegler catalysts[6]. Since then, the role of nickel has changed drastically after decades of research. Currently, a large number of Ni(II)-based catalysts have been reported with the ability to generate high molecular weight polyolefins. Brookhart's seminal works in 1995 showed that α-diimine nickel catalysts (Fig. 1, **I**) were capable of producing high molecular weight polyethylene with activities rivaling many early transition metal catalysts[7–9]. The salicylaldimine nickel catalysts (**II**) developed by Grubbs and co-workers in 2000 represent another notable advance in this field[10–14]. SHOP-type nickel catalysts (**III**; SHOP = Shell Higher Olefin Process) have been commercialized for the synthesis of linear α-olefins[15]. With proper structural modifications, this type of catalyst can produce high molecular weight polyethylene[16]. For example, some nickel complexes bearing o-bis(aryl)-phosphino-phenolate ligands (**IV**) possess advantageous properties for ethylene polymerization and copolymerization with acrylate comonomers[17,18]. Recently, extensive research efforts have been focused on the design and development of nickel catalysts based on phosphine-derived ligands (**V–VII**)[19–22], partially due to their potential for generating polar-functionalized polyolefins through copolymerization.

Despite recent research interest in phosphine-related ligands, imine-derived ligands are attractive platforms due to their many unique properties (e.g., easy to synthesize and modify). Moreover, phosphine-type ligands possess some drawbacks, such as high oxygen sensitivity as well as potential pyrophoricity and toxicity, that require special handling[23]. Interestingly, our literature survey indicates that the phosphine-based nickel catalysts (**III–VII**) typically lead to the formation of highly linear polyethylenes in ethylene polymerization[24,25]. In contrast, most N^N-based or N^O-based nickel catalysts (**I**, **II**, and many others) generate variously branched polyethylenes[26–32]. Industrially, branches are introduced through α-olefin copolymerization to improve the processability as well as many other material properties of polyolefins. The ability of these nitrogen-based nickel catalysts to produce branched polyethylenes using only ethylene as a feedstock is highly attractive from an industrial standpoint and has indeed led to extensive research efforts from large chemical companies, such as DuPont[33]. Based on these considerations as well as our ongoing efforts towards developing high-performance

nickel catalysts, we became interested in the α-imino-ketone ligand framework. Brookhart and co-workers briefly studied this type of ligand in a patent in 2000[34]. However, only moderate activity ($\sim 6.4 \times 10^4$ g mol$^{-1}$ h$^{-1}$ at room temperature and 6.9 MPa of ethylene pressure) and moderate polyethylene molecular weights ($M_n < 80,000$) were reported, and no studies on the influence of ligand structures or polymerization conditions were performed.

The synthesis of extensively studied α-diimine ligands sometimes requires harsh reaction conditions along with very low yields. For example, Long and coworkers recently described the synthesis of sterically demanding Ni(II) α-diimine complex utilizing 2,6-bis(diphenyl-methyl)−4-methyl aniline, which is capable of producing polyethylene at temperatures up to 100 °C. However, very low yield (8.6%) was reported for the α-diimine ligand synthesis[35]. They subsequently reported a modified synthetic procedure for an acenaphthenequinone derived ligand, which still only led to 10% yield[36]. As a matter of fact, α-imino-ketone ligands are the key intermediate/side products during the synthesis of various α-diimine ligands. It is somewhat surprising that they have not been extensively explored already. During our studies, it was observed that the reaction of α-imino-ketone ligand with (DME)NiBr$_2$ (DME = ethylene glycol dimethyl ether) led to unstable nickel species. The addition of aluminum cocatalyst led to catalyst decomposition and no activity in ethylene polymerization. Comparing with α-diimine ligand, α-imino-ketone ligand is electronically less donating and sterically less bulky, leading to a more nucleophilic and more sterically open metal center, which are beneficial for olefin monomer access and insertion. Unfortunately, these features also make it incapable of stabilizing a neutral nickel center (NiBr$_2$ for example). In this system, a pre-generated cationic nickel center ([Ni(allyl)]$^+$) was employed to address this issue. It should be noted that nickel allyl initiators are known in ethylene polymerization, but did not receive much attention[37,38]. Most studies using neutral ligands such as α-diimine or pyridine-imine[26,27] involved the synthesis of neutral (ligand)NiBr$_2$ complexes, and the combination with aluminum cocatalyst generates active catalyst for olefin polymerization. Our preliminary studies showed that the combination of α-diimine or pyridine-imine ligands with the pre-generated cationic nickel center ([Ni(allyl)]$^+$) is also highly active in ethylene polymerization. We envision that this nickel allyl route provides an easy to handle, cheap and safe alternative versus the abovementioned traditional strategy.

In this contribution, we wish to demonstrate that α-imino-ketone nickel is a highly versatile platform with many unique properties as follows: they are extremely easy to synthesize and

**Fig. 1 Selected examples of nickel-based ethylene polymerization catalysts.** Various high-performance ethylene (co)polymerization systems have been reported, with notable examples including α-diimine nickel catalysts (**I**), salicylaldimine nickel catalysts (**II**), SHOP-type nickel catalysts (**III**), phosphinophenolate nickel catalysts (**IV**), phosphine-sulfonate nickel catalysts (**V**), diphosphazane monoxide nickel catalysts (**VI**), and triadamantylphosphine nickel catalysts (**VII**). In this work, a simple but extremely versatile α-imino-ketone nickel system was reported.

**Fig. 2 Synthesis of α-imino-ketone ligands and the corresponding nickel complexes.** Reactions of diketone with anilines form α-imino-ketone ligands (**L1**–**L7**). Reactions the ligands with [Ni(allyl)Cl]$_2$ and NaBAF form the desired cationic nickel complexes (**Ni1**–**Ni7**).

modify, making them potentially amenable for high-throughput screening experiments[39]; their single-component nature avoids the use of expensive and pyrophoric aluminum cocatalyst, which also eliminates chain transfer to the cocatalyst, to yield high molecular weight polymers and copolymers even at high polymerization temperatures; despite the sterically open nature, this class of nickel catalyst enables access to ultra-high-molecular-weight polyethylene (UHMWPE, molecular weights: $10^6$–$10^7$), which is an important specialty class of polyethylene with wide applications[40]; and the versatility of this ligand platform allows facile synthesis of dinuclear counterpart and the installation of an anchor for catalyst heterogenization.

## Results

**Synthesis and characterization of the nickel complexes.** The α-imino-ketone ligands can be easily prepared from the reactions of diketone with anilines at high yields (Fig. 2). In most cases, both reactants are commercially available or easily synthesized through simple reactions. The ligand framework is highly versatile with many positions that can be independently modified. Subsequently, the cationic nickel complexes (**Ni1**–**Ni7**) were obtained from the reactions of the ligands (**L1**–**L7**) with [Ni(allyl)Cl]$_2$ and sodium tetrakis(3,5-bis(trifluoromethyl)phenyl)borate (NaBAF). These ligands and metal complexes were characterized by $^1$H nuclear magnetic resonance (NMR), $^{13}$C NMR, mass spectrometry and elemental analysis.

The molecular structures of **Ni1** and **Ni6** were determined by X-ray diffraction analysis (Fig. 3). The geometry at the nickel center is square planar, making these nickel complexes diamagnetic and suitable for NMR characterization. A distinguishing feature of this system is the sterically open configuration that allows one side of the nickel center to be completely exposed, which is in direct contrast to the extensively studied Brookhart type α-diimine nickel system. This configuration can lead to better accessibility of the nickel center, which may in turn increase catalytic activity as well as increase comonomer incorporation in copolymerization reactions. The calculated topographical steric map analyses[41] confirmed that **Ni1** is sterically much more open

than an analogous α-diimine-based nickel complex[42] (buried volume %$V_{Bur}$ of 39.9 vs. 46.6, Supplementary Figs. 1, 2).

**Ethylene polymerization studies.** These nickel complexes are highly active in ethylene polymerization without the addition of any cocatalyst (Table 1). These nickel complexes demonstrated activities close to and more than $10^8$ g mol$^{-1}$ h$^{-1}$. In comparison to the *iso*-propyl-substituted catalyst **Ni1**, **Ni2** bearing a sterically bulkier diphenyl-methyl substituent led to the formation of polyethylene with up to 9 times higher molecular weights (Table 1, entries 1–6). Most importantly, **Ni2** is able to generate high molecular weight polyethylene even at 80 °C. Time dependence studies showed that **Ni1** remained active for 30 min at 80 °C, and the thermal stability is comparable with the α-diimine counterpart (Supplementary Fig. 3). The polymer molecular weight remained constant as the ethylene pressure increased (Table 1, entries 7–9). This result indicates that chain transfer to monomer is the predominant chain transfer pathway. Therefore, the propagation/chain transfer ratio is independent of the ethylene pressure[43]. Interestingly, the polymer branching density is largely independent of the ethylene pressure. In the α-diimine nickel system, the polymer molecular weight increased significantly along with a significantly decreased polymer branching density as the ethylene pressure increased[44]. This contrast may originate from the sterically open nature of the α-imino-ketone system compared to the α-diimine system.

The in situ mixing of the ligands with the nickel precursors (without isolation/purification of the nickel complex) led to the same polymerization properties compared to the discrete metal complexes. By changing the polymerization solvent from toluene to an industrially preferred hydrocarbon solvent, increased activity ($1.06 \times 10^8$ g mol$^{-1}$ h$^{-1}$) along with a similar molecular weight were observed (Table 1, entry 10). These attractive features make this system potentially amenable to high-throughput screening experiments as well as continuous polymerization processes.

The utilization of substituted naphthalenyl amines[45] led to reduced catalytic activities (**Ni3** and **Ni4**). However, greatly increased polymer molecular weights were achieved. Specifically,

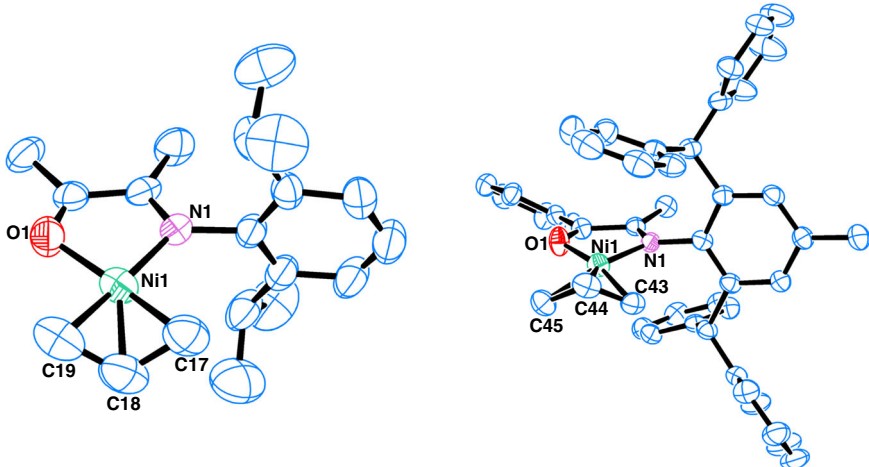

**Fig. 3 Molecular structures of Ni1 and Ni6.** Hydrogen atoms and BAF group were omitted for clarity. Selected bond lengths (Å) and angles (°): Ni1–N1 = 1.925(3), Ni1–O1 = 1.946(3), Ni1–C17 = 1.98(8), Ni1–C18 = 1.916(18), Ni1–C19 = 1.95(7), N1–Ni1–O1 = 82.05(13), C19–Ni1–C17 = 74(3); Ni1–C44 = 1.921 (14), Ni1–O1 = 1.923(4), Ni1–N1 = 1.942(4), Ni1–C43 = 2.01(5), Ni1–C45 = 2.03(2), O1–Ni1–N1 = 81.29(16), C43–Ni1–C45 = 72.3(17).

### Table 1 Ethylene homopolymerization with Ni complexes.

| Entry | Catalyst | Temperature (°C) | Pressure (atm) | Yield (g) | Act.[a] | $M_n$[b] (10⁴) | PDI[b] | B[c] | $T_m$ (°C)[d] |
|---|---|---|---|---|---|---|---|---|---|
| 1 | Ni1 | 20 | 8 | 1.0 | 2.0 | 8.6 | 3.1 | 10 | 129.7 |
| 2 | Ni1 | 50 | 8 | 1.5 | 3.0 | 3.6 | 2.6 | 19 | 123.5 |
| 3 | Ni1 | 80 | 8 | 1.4 | 2.8 | 3.1 | 2.3 | 23 | 117.2 |
| 4 | Ni2 | 20 | 8 | 0.9 | 1.8 | 28.9 | 3.2 | 13 | 126.3 |
| 5 | Ni2 | 50 | 8 | 1.8 | 3.6 | 25.7 | 1.9 | 17 | 125.4 |
| 6 | Ni2 | 80 | 8 | 1.1 | 2.2 | 23.9 | 2.0 | 20 | 114.2 |
| 7 | Ni2 | 50 | 15 | 2.8 | 5.6 | 24.5 | 2.3 | 14 | 123.1 |
| 8 | Ni2 | 50 | 20 | 3.0 | 6.0 | 24.1 | 2.9 | 13 | 123.9 |
| 9 | Ni2 | 50 | 25 | 4.2 | 8.4 | 25.8 | 2.5 | 12 | 125.1 |
| 10[e] | Ni2 | 50 | 25 | 5.3 | 10.6 | 25.2 | 2.4 | 13 | 127.5 |
| 11[f] | Ni3 | 20 | 8 | 1.3 | 0.13 | 57.3 | 3.0 | 23 | 109.3 |
| 12[f] | Ni3 | 50 | 8 | 0.9 | 0.09 | 21.4 | 2.0 | 32 | 100.6 |
| 13[f] | Ni3 | 80 | 8 | 0.7 | 0.07 | 3.97 | 2.0 | 67 | 57.8 |
| 14[f] | Ni4 | 20 | 8 | 1.4 | 0.14 | 26.3 | 2.2 | 57 | 112.8 |
| 15[f] | Ni4 | 50 | 8 | 0.8 | 0.08 | 46.1 | 2.1 | 61 | 82.8 |
| 16[f] | Ni4 | 80 | 8 | 0.6 | 0.06 | 122.6 | 2.0 | 70 | 63.4 |

Polymerization conditions: catalyst = 1 μmol; toluene = 28 mL, $CH_2Cl_2$ = 2 mL, time = 3 min
[a]The yields and activities are averages of at least two runs. The unit for activity is defined as $10^7$ g mol⁻¹ h⁻¹
[b]Determined by GPC in trichlorobenzene at 150 °C
[c]B = branches per 1000 carbons, determined by ¹H NMR spectroscopy
[d]Determined by differential scanning calorimetry (DSC)
[e]Heptane = 28 mL, $CH_2Cl_2$ = 2 mL
[f]Time = 60 min

polyethylene with a molecular weight of more than one million can be generated by **Ni4** at 80 °C. Interestingly, **Ni3/Ni4** afforded polyethylenes with much higher branching densities (23–70/1000 C) than those obtained from **Ni1/Ni2**. The electronic effect of the ketone side was investigated using complexes **Ni5–Ni7** (Supplementary Table 1). Complexes **Ni5–Ni7** exhibited high activities (1.2–2.6 × $10^7$ g mol⁻¹ h⁻¹) that are comparable to those of **Ni1/Ni2**. In addition, polyethylenes with very high molecular weights ($M_n$ up to 8.19 × $10^5$) were generated even at high temperatures. Specifically, complex **Ni7** bearing an electron-withdrawing substituent (i.e., $NO_2$) resulted in greatly reduced polymer branching densities (5–8/1000 C) and increased melting points (124.7–129.5 °C) compared to those of **Ni5** and **Ni6** (12–23/1000 C and 110.8–122.8 °C, respectively).

**Ethylene-polar monomer copolymerization studies.** Nickel catalysts based on α-diimine ligands can enable copolymerization

reaction of ethylene with methyl acrylate (MA)[46]. However, very harsh polymerization conditions (120 °C and 1000 psi ethylene pressure) and a large amount of tris(pentafluorophenyl)borane additive were required. In this system, moderate activity and copolymer molecular weight were achieved under mild conditions without the addition of any cocatalyst or protecting agent (Table 2, entry 1). Very high activity and high incorporation (10.8%) were observed with trimethoxyvinylsilane (Table 2, entry 2), indicating the low poisoning effect of this type of comonomer[47,48]. 5-norbornene-2-yl acetate is also suitable comonomer for copolymerization (Table 2, entry 3).

This catalytic system is particularly suitable for ethylene copolymerization with special comonomers bearing long spacers between the double bond and the polar groups. This system demonstrated high activities (up to 2.5 × $10^5$ g mol⁻¹ h⁻¹), very high copolymer molecular weights ($M_n$ up to 306,700) and high comonomer incorporation (1.8–6.3%) towards comonomers

**Table 2 Ethylene copolymerizations with polar monomers with Ni complexes.**

| Entry | Catalyst | Comonomer | Temperature (°C) | [M] mol/L | Yield (g) | Act.[b] | $X_M$[c] (%) | $M_n$[d] | PDI[d] | $T_m$ (°C)[e] |
|---|---|---|---|---|---|---|---|---|---|---|
| 1 | Ni2 | COOMe | 50 | 0.5 | 0.1 | 0.25 | 0.6 | 17900 | 2.0 | 124.9 |
| 2 | Ni2 | Si(OMe)₃ | 50 | 1 | 8.8 | 22 | 10.8 | 8300 | 5.4 | 104.0 |
| 3 | Ni2 | (cyclic ester) | 50 | 1 | 0.3 | 0.8 | 5.5 | 29000 | 1.7 | 96.2 |
| 4 | Ni2 | ⸀⁄₄Cl | 50 | 1 | 2.3 | 5.8 | 2.2 | 306700 | 2.4 | 109.3 |
| 5 | Ni2 | ⸀⁄₄Cl | 50 | 2 | 2.0 | 5.0 | 3.4 | 151800 | 1.8 | 102.5 |
| 6 | Ni2 | ⸀⁄₈COOMe | 20 | 1 | 4.0 | 10 | 2.8 | 72200 | 1.6 | 109.8 |
| 7 | Ni3 | ⸀⁄₈COOMe | 20 | 1 | 2.0 | 5.0 | 2.8 | 55500 | 1.6 | 109.6 |
| 8 | Ni4 | ⸀⁄₈COOMe | 20 | 1 | 5.3 | 13.3 | 1.8 | 165200 | 1.8 | 95.9 |
| 9 | Ni2 | ⸀⁄₈COOMe | 50 | 1 | 10 | 25 | 3.2 | 50100 | 2.0 | 110.6 |
| 10 | Ni2 | ⸀⁄₈COOMe | 50 | 2 | 7.0 | 17.5 | 6.3 | 33300 | 1.8 | 95.0 |

Polymerization conditions: total volume of toluene and polar monomer = 20 mL, catalyst = 20 μmol, time = 2 h
[a]The yields and activities are averages of at least two runs. The unit for activity is defined as $10^4$ g mol$^{-1}$ h$^{-1}$
[b]Determined by [1]H NMR integration in $C_2D_2Cl_4$ at 120 °C
[c]Determined by GPC in trichlorobenzene at 150 °C
[d]Determined by DSC

6-chloro-1-hexene and methyl 10-undecenoate (Table 2, entries 4–10). Methyl 10-undecenoate has the added advantage of being biorenewable and readily available from cross metathesis using fatty acid derivatives, such as methyl oleate[49]. These superior features make this approach an efficient route to access polar-functionalized linear low-density polyethylene (LLDPE)[50]. In previously reported ethylene-polar monomer copolymerization studies, a high comonomer concentration is required to achieve high incorporation, which leads to a very low comonomer utilization ratio. The most distinguishing feature of this system is the high comonomer utilization ratio (47.8% and 28.5% of methyl 10-undecenoate were consumed for entries 9 and 10, respectively). This ratio can be further improved through optimization of catalyst structures and polymerization conditions. Consistent with the above-mentioned studies, this unique feature is probably due to the sterically open nature of the α-imino-ketone system.

**Dinuclear nickel complexes**. The metal–metal cooperativity effect has been demonstrated to be a powerful strategy to mediate the olefin polymerization processes[51]. However, the design and synthesis of dinuclear metal complexes can be complicated and time consuming. Due to the structural simplicity of the α-imino-ketone platform, dinucleating ligands can be easily prepared starting from various di-amines. Here, a dinuclear nickel complex (Fig. 4, **Ni–Ni**) was prepared using a xanthene-bridged di-amine[52,53]. In comparison to the mono-nuclear analog (**Ni1**), **Ni–Ni** exhibited similar activities, increased polymer molecular weights and reduced polyethylene branching densities (Table 3, entries 1–3).

**Heterogeneous nickel complex**. Heterogeneous olefin poly-merization catalysts are preferred from an industrial perspective due to their ability to control product morphology, preventing reactor fouling and enabling continuous polymerization pro-cess[54]. The heterogenization of early transition metal catalysts has been extensively studied, and many systems have been success-fully commercialized[55,56]. In comparison, very few studies on the heterogenization of late transition metal catalysts have been

**Fig. 4 Chemical structures of the dinuclear and supported nickel complexes.** A dinuclear nickel complex (**Ni**-**Ni**) was prepared from a xanthene-bridged di-amine. A SiO₂ supported nickel complex (**Ni**-**OH@SiO₂**) was generated from the interaction of a pre-installed anchoring group (OH) with the solid support.

reported[57–61]. A homogeneous catalyst supported on a solid support can lead to serious steric crowding and complications due to side reactions, which could greatly reduce catalytic activities[62]. In some cases, the support of homogeneous catalyst would limit monomer access to the metal center, thereby decreasing poly-merization activity. Due to the single-component and sterically open characteristics of this type of catalysts, we envision that a straightforward process for heterogenization on a SiO₂ support. Unfortunately, these nickel catalysts exhibited very low affinity for the SiO₂ support despite their cationic nature. In most previous heterogenization studies, the SiO₂ support must be pretreated with an aluminum type of cocatalyst, such as methy-laluminoxane, to increase the affinity of the metal complex for the solid support[62,63]. Furthermore, a large amount of additional cocatalyst is required to achieve high catalytic activity. In this study, we utilized an alternative strategy to address this issue through the installation of an anchoring group (OH) onto the catalyst. Indeed, the corresponding **Ni–OH** complex can be easily supported on SiO₂, leading to a highly active heterogeneous

**Table 3 Ethylene homopolymerization with Ni complexes.**

| Entry | Catalyst | Temperature (°C) | Pressure (atm) | Yield (g) | Act.[a] | $M_n$[b] ($10^4$) | PDI[b] | B[c] | $T_m$(°C)[d] |
|---|---|---|---|---|---|---|---|---|---|
| 1 | Ni–Ni | 20 | 8 | 0.7 | 1.4 | 12.9 | 2.0 | 8 | 131.1 |
| 2 | Ni–Ni | 50 | 8 | 1.0 | 2.0 | 4.7 | 1.7 | 16 | 123.6 |
| 3 | Ni–Ni | 80 | 8 | 0.8 | 1.6 | 4.0 | 1.8 | 22 | 119.8 |
| 4 | Ni–OH | 20 | 8 | 0.8 | 1.6 | 104.3 | 1.8 | 15 | 124.6 |
| 5 | Ni–OH | 50 | 8 | 0.7 | 1.4 | 54.4 | 2.4 | 16 | 116.6 |
| 6 | Ni–OH | 80 | 8 | 0.6 | 1.2 | 48.2 | 2.4 | 16 | 114.3 |
| 7 | Ni–OH | 100 | 8 | 0.35 | 0.7 | 23.4 | 2.4 | 18 | 113.9 |
| 8[e] | Ni–OH | 100 | 8 | 0.5 | 0.1 | 22.6 | 2.3 | 17 | 115.6 |
| 9[f] | Ni–OH@SiO₂ | 50 | 8 | 2.2 | 0.44 | 155.6 | 2.7 | 10 | 129.9 |
| 10[f] | Ni–OH@SiO₂ | 80 | 8 | 1.1 | 0.22 | 72.2 | 3.0 | 11 | 126.4 |
| 11[f] | Ni–OH@SiO₂ | 100 | 8 | 0.8 | 0.16 | 70.1 | 2.9 | 16 | 122.7 |
| 12[f] | Ni–OH@SiO₂ | 50 | 15 | 2.8 | 0.56 | 159.3 | 3.0 | 11 | 131.1 |
| 13[f] | Ni–OH@SiO₂ | 50 | 20 | 3.2 | 0.64 | 146.5 | 2.9 | 10 | 131.1 |
| 14[f] | Ni–OH@SiO₂ | 50 | 25 | 4.0 | 0.8 | 144.4 | 2.4 | 9 | 131.9 |

Polymerization conditions: catalyst = 1 μmol; toluene = 28 mL, CH₂Cl₂ = 2 mL, time = 3 min
[a]The yields and activities are average of at least two runs. The unit for activity is defined as $10^7$ g mol⁻¹ h⁻¹
[b]Determined by GPC in trichlorobenzene at 150 °C
[c]B = branches per 1000 carbons, determined by ¹H NMR spectroscopy
[d]Determined by differential scanning calorimetry (DSC)
[e]Time = 30 min
[f]Time = 30 min, Heptane = 30 mL

catalyst (Fig. 4). Using this approach, pretreatment of the solid support or use of any cocatalyst can be avoided.

Homogeneous complex **Ni–OH** exhibited very high activities ($1.6 \times 10^7$ g mol⁻¹ h⁻¹) for ethylene polymerization, generating polyethylene with molecular weights of more than one million (Table 3, entries 4–8). The ability of complexes **Ni4** and **Ni–OH** to generate UHMWPE is quite surprising considering their sterically open characteristics. The synthesis of UHMWPE using nickel catalysts remains challenging, and very few nickel systems have these capabilities[64–70]. Furthermore, it is even more challenging to develop heterogeneous nickel catalyst with such capabilities, and to the best of our knowledge, this type of system has not been reported previously.

The supported catalyst **Ni–OH@SiO₂** exhibited greatly increased polymerization activity compared to its homogeneous counterpart. In addition, polyethylene with substantially enhanced molecular weights and slightly decreased branching densities were generated (Table 3, entries 9–11). The polymerization activities slightly increased with increasing ethylene pressure (Table 3, entries 12–14). In addition, the polyethylene molecular weights and polymer branching densities remained constant, which is similar to the results for homogeneous catalyst **Ni2**. **Ni–OH@SiO₂** is able to generate polyethylene with molecular weights of much higher than one million ($M_n$ up to $1.593 \times 10^6$). Most surprisingly, high molecular weight polyethylene ($M_n = 7.01 \times 10^5$) can be obtained even at 100 °C, demonstrating the superior properties of heterogeneous catalyst **Ni–OH@SiO₂**. Interestingly, **Ni–OH@SiO₂** generated polyethylenes with more linear microstructure and higher melting points compared to those of the homogeneous counterpart **Ni–OH**. Probably, the interaction of the OH moiety on the ligand backbone with the SiO₂ surface alters the electronic environment around the nickel center, influencing the polymerization properties.

The results from time dependent studies indicate that heterogeneous catalyst **Ni–OH@SiO₂** is thermally much more stable than the homogeneous counterpart **Ni–OH** (Fig. 5, Supplementary Table 2). Significant decomposition was observed for **Ni–OH** after 3 min at 100 °C (Fig. 5; Table 3, entries 7 and 8). In contrast, **Ni–OH@SiO₂** remained highly active within 2 h at 100 °C. More importantly, **Ni–OH@SiO₂** led to the formation of while powders while **Ni–OH** led to greenish products, clearly indicating catalyst decomposition (Supplementary Fig. 4).

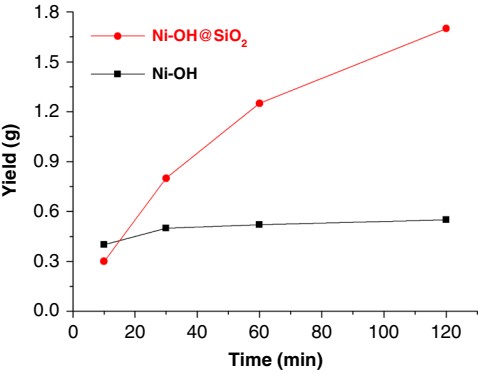

**Fig. 5 Polymer yield as a function of time for Ni-OH@SiO₂ and Ni-OH.** **Ni**-**OH@SiO₂** is thermally more stable than **Ni**-**OH** in ethylene polymerization at 100 °C, and remained highly active within 120 min (homogeneous polymerization conditions: ethylene = 8 atm, catalyst = 1 μmol, heptane = 29 mL, DCM = 1 mL; heterogeneous polymerization conditions: ethylene = 8 atm, catalyst = 1 μmol catalyst supported on 100 mg of SiO₂, heptane = 30 mL).

Furthermore, **Ni–OH** led to the formation of an agglomerated polymer product that sticks to the polymerization vessel. However, **Ni–OH@SiO₂** yielded free-flowing polymer particles (Supplementary Fig. 5). The polymer particles increase in size along with polymerization time, and replicate the shape of the starting silica particles (Supplementary Fig. 6). These superior features make the heterogeneous system highly attractive and suitable for an industrially preferred continuous process.

## Discussion

A simple but versatile α-imino-ketone nickel system has been demonstrated to exhibit great properties for ethylene polymerization and copolymerization with polar monomers. Despite their sterically open nature, these nickel catalysts are thermally stable with the ability to generate polymers with very high molecular weights and tunable branching densities. To further demonstrate the versatility of this system, a dinucleating ligand was designed to utilize the metal–metal cooperativity effect and

an anchoring moiety was introduced to facilitate heterogenization of the homogeneous nickel catalyst. This system combines many superior properties, such as ease of synthesis, single component nature, high thermal stability, generation of high molecular weight polymers and simplicity in heterogenization and modification, making it highly attractive for future investigations.

## Methods

**Measurements.** All experiments were carried out under dry Nitrogen atmosphere using standard Schlenk techniques or in a glove-box. Deuterated solvents used for NMR were dried and distilled prior to use. $^1H$, $^{13}C$ NMR spectra were recorded a Bruker Ascend$^{Tm}$ 400 spectrometer at ambient temperature unless otherwise stated. The chemical shifts of the $^1H$ and $^{13}C$ NMR spectra were referenced to tetramethylsilane. Coupling constants are in Hz. Elemental analysis was performed by the Analytical Center of the University of Science and Technology of China. X-ray diffraction data were collected at 298 (2) K on a Bruker Smart CCD area detector with graphite-monochromated Mo Kα radiation ($\lambda = 0.71073$ Å). Molecular weights and molecular weight distributions of the polymers were determined by gel permeation chromatography (GPC) using an Agilent PL-220 chromatograph equipped with two Agilent PLgel Olexis columns operating at 150 °C using o-dichlorobenzene as the solvent. The system was calibrated with a polystyrene standard, and chromatograms were corrected for linear polyethylene through universal calibration using the Mark–Houwink parameters of Rudin: $K = 1.75 \times 10^{-2}$ cm$^3$/g and $R = 0.67$ for polystyrene, and $K = 5.90 \times 10^{-2}$ cm$^3$/g and $R = 0.69$ for polyethylene. Dichloromethane, hexanes, and toluene were purified by solvent purification systems. 955 SiO$_2$ was purchased from Grace Davison Co., Ltd. (average pore diameter = 22.6 nm, surface area = 264 m$^2$/g, average particle size = 40 μm). The silica was activated at 600 °C for 5 h before use.

**Procedure for ethylene homopolymerization.** In a typical experiment, a 350 mL pressure vessel was charged with 28 mL toluene or 28 mL heptane, the catalyst sample (supported nickel complex or nickel complex solution in dichloromethane) and a magnetic stir bar in the glovebox. The pressure vessel was connected to a high pressure line and the solution was degassed. The vessel was warmed to setting temperature using an oil bath (water bath for the case of polymerization at room temperature) and allowed to equilibrate for 15 min. With rapid stirring, the reactor was pressurized and maintained at 8.0 atm of ethylene. After desired amount of time, the pressure vessel was vented and the polymer was precipitated in methanol and dried at 50 °C for 24 h under vacuum.

**Procedure for ethylene-polar monomer copolymerization.** In a typical experiment, a 350 mL pressure vessel was charged with toluene and polar monomer in total 18 mL and a magnetic stir bar in the glovebox. The pressure vessel was connected to a high pressure line and the solution was degassed. The vessel was warmed to 50 °C using an oil bath (water bath for the case of polymerization at room temperature) and allowed to equilibrate for 15 min. Twenty micromolar of Ni complex in 2 mL CH$_2$Cl$_2$ was injected into the polymerization system via syringe. With rapid stirring, the reactor was pressurized and maintained at 8.0 atm of ethylene. After 2 h, the pressure vessel was vented and the polymer was precipitated in methanol and dried at 50 °C for 24 h under vacuum.

## Data availability

The X-ray crystallographic data for complex **Ni1** has been deposited at the Cambridge Crystallographic Data Center (CCDC) under the deposition number 1828966. The X-ray crystallographic data for complex **Ni6** has been deposited at the Cambridge Crystallographic Data Center (CCDC) under the deposition number 1814000. These data can be obtained free of charge via www.ccdc.cam.ac.uk/data_request/cif. Supplementary Information contains detailed experimental procedures and characterization data for new compounds ((Supplementary Methods), as well as DSC (Supplementary Figs. 103-151), GPC (Supplementary Figs. 152–200), X-ray crystallography (Supplementary Table 3 and Supplementary Table 4), $^{13}C$ and $^1H$ NMR data (Supplementary Figs. 7–46 and Supplementary Figs. 54–102), ESI-MS (Supplementary Figs. 47–53). All data can be supplied by the authors upon reasonable request.

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

## Acknowledgements

This work was supported by National Natural Science Foundation of China (NSFC, 21690071 and 21871242), the Fundamental Research Funds for the Central Universities.

## Author contributions

T.L. and C.C.L. conceived the study and planned the research. T.L. and S.B.G. conducted the catalysts synthesis and olefin polymerization experiments. C.C.L. prepared the manuscript.

## Competing interests

The authors declare no competing interests.
