## [Peer Review File · Nature Communications]

REVIEWERS' COMMENTS:

Reviewer #1 (Remarks to the Author):

The authors have significantly modified the manuscript based on the reviewer comments. The unique features of the reported catalyst system vs. well-studied alfa-diimine Ni catalyst systems are now more clearly defined. This ambitious manuscript is now supported with more solid details. It has been getting increasingly challenging to develop new Ni catalyst systems and polymerization methodologies even though the goals are still very important. In this regard, this is a nice piece of work. Thus, this reviewer recommends publication in Nature Commun with no further revisions.

Reviewer #2 (Remarks to the Author):

The authors have addressed the comments from Reviewer 1 and myself, and I recommend this manuscript for publication.

Reviewer #1 (Remarks to the Author):

The authors have significantly modified the manuscript based on the reviewer comments. The unique features of the reported catalyst system vs. well-studied alfa-diimine Ni catalyst systems are now more clearly defined. This ambitious manuscript is now supported with more solid details. It has been getting increasingly challenging to develop new Ni catalyst systems and polymerization methodologies even though the goals are still very important. In this regard, this is a nice piece of work. Thus, this reviewer recommends publication in Nature Commun with no further revisions.

Answer: Thanks a lot for your comments.

Reviewer #2 (Remarks to the Author):

The authors have addressed the comments from Reviewer 1 and myself, and I recommend this manuscript for publication.

Answer: Thanks a lot for your comments.